# Epidemiology and molecular characterization of *Staphylococcus aureus* causing bovine mastitis in water buffaloes from the Hazara division of Khyber Pakhtunkhwa, Pakistan

**Salma Javed**[1], **JoAnn McClure**[2], **Muhammad Ali Syed**[3]*, **Osahon Obasuyi**[4], **Shahzad Ali**[5], **Sadia Tabassum**[1], **Mohammad Ejaz**[3], **Kunyan Zhang**[2,4,6,7,8]*

**1** Department of Zoology, Hazara University, Mansehra, Pakistan, **2** Centre for Antimicrobial Resistance, Alberta Health Services/Alberta Precision Laboratories/University of Calgary, Calgary, Alberta, Canada, **3** Department of Microbiology, The University of Haripur, Haripur, Pakistan, **4** Department of Pathology & Laboratory Medicine, University of Calgary, Calgary, Alberta, Canada, **5** Department of Wildlife and Ecology, One Health Research Group, Discipline of Zoology, University of Veterinary and Animal Sciences, Lahore, Pakistan, **6** Department of Microbiology, Immunology and Infectious Diseases, University of Calgary, Calgary, Alberta, Canada, **7** Department of Medicine, University of Calgary, Calgary, Alberta, Canada, **8** The Calvin, Phoebe and Joan Snyder Institute for Chronic Diseases, University of Calgary, Calgary, Alberta, Canada

* syedali@uoh.edu.pk (MAS); kzhang@ucalgary.ca (KZ)

**Data Availability Statement:** All relevant data are within the paper and its Supporting Information files.

## Abstract

Buffalo represent a major source of milk in Pakistan. However, production is impacted by the disease bovine mastitis. Mastitis causes significant economic losses, with *Staphylococcus aureus* (*S. aureus*) being one of its major causative agents. While much work has been done understanding the epidemiology of bovine mastitis in Pakistan, detailed molecular characterization of the associated *S. aureus* is unavailable. In the current study both the epidemiological and molecular characterization of *S. aureus* from bovine mastitis in the Hazara division of Pakistan are examined. *S. aureus* was isolated from 18.41% of the animals, and left quarters more prone to infection (69.6%) than right quarters (30.4%). Sub-clinical mastitis (75.31%) was more prevalent than clinical mastitis (24.69%), with infections evenly distributed amongst the eight districts. Molecular characterization revealed that only 19.6% of the isolates were methicillin-resistant, and four strains types identified, including ST9-t7867-MSSA, ST9-MSSA, ST101-t2078-MSSA, and ST22-t8934-MRSA-IVa. Antiseptic resistance genes were not detected in the isolates, and low levels of antibiotic resistance were also noted, however the methicillin-resistant strains had higher overall antibiotic resistance. This study represents the most complete molecular typing data for *S. aureus* causing bovine mastitis in the Hazara district of Pakistan, and the country as a whole.

## Introduction

Buffalo are the most frequently reared animals in Asia, estimated to account for 96.78% of the world's buffalo milk production, at 89.2 million tons [1]. In Pakistan, buffalo produce 68.4% of the country's milk, however, production is threatened by the infectious disease bovine mastitis

**Funding:** KZ: This work was supported in part by the operating fund from the Centre for Antimicrobial Resistance (CAR), Alberta Health Services, Alberta, Canada. The funders had no role in study design, data collection and analysis, decision to publish, or preparation of the manuscript.

**Competing interests:** The authors have declared that no competing interests exist.

[1]. Mastitis in the udders of dairy cattle occurs when pathogenic bacteria enter through the teat canal, where they colonize and multiply in the alveoli. The bacteria produce toxic substances that cause damage to the milk producing tissues, resulting in reduced milk yield and quality [2]. Mastitis can occur in two forms; clinical and subclinical. In clinical mastitis clots and flakes can be seen in the milk and the quarters become swollen, with severe conditions leading to the formation of lacerations, necrosis and cord formation of the teat [3]. In subclinical mastitis (SCLM) no clinical signs or symptoms are apparent, though there is a reduction in milk quantity and quality. Subclinical infection is 15–40 times more prevalent than clinical infection and persists for a long time, rapidly spreading through a herd [4].

Mastitis is one of the leading diseases of the dairy industry, causing economic loss due to low milk yield or quality, loss of lactation, poor animal health, premature culling, and wastage of milk due to over use of antibiotics [5]. Mastitis is two times more prevalent in hand-milked (25.1%) vs machine milked (14.6%) animals [6] and in most developing countries, including Pakistan, dairy animals are milked by hand. Based on mode of transmission and primary reservoir, mastitis pathogens are classified into environmental pathogens that enter into host body from environment and the contagious pathogens are those that transfer from one animal to another. *Staphylococcus aureus* is a contagious pathogen that efficiently adapts to the environmental conditions of the mammary glands and spreads to and between animals during milking. Infection with *S. aureus* causes an inflammatory reaction that can result in tissue damage and acute infection can lead to gangrenous mastitis, characterized by blue to black quarters that eventually slough off, often associated with the production of staphylococcal alpha toxin [1].

In Pakistan, *S. aureus* is one of the most common causative agents of bovine mastitis in buffaloes, accounting for 46.72% of cases [7]. While numerous studies have examined the prevalence and epidemiology of mastitis in various regions of Pakistan, detailed molecular typing of the staphylococci from cases of mastitis in the country is limited [8–11]. The present study, therefore, examines both the epidemiology and molecular characterization of *S. aureus* from bovine mastitis cases in water buffaloes of different districts of Hazara Division, Khyber Pakhtunkhwa (KP), Pakistan.

## Materials and methods

### Study area and data collection

The study was conducted in the Hazara Division of KP, Pakistan (August 2019-January 2020). KP is one of the five administrative provinces of Pakistan, located in the North-Western part of the country, across the international border with Afghanistan. The province of Khyber Pakhtunkhwa is divided into 35 districts and seven divisions, with Hazara being one of the seven Divisions, itself containing eight districts; Haripur, Abbottabad, Mansehra, Battagram, Kohistan Upper, Kohistan Lower, Torghar and Kolai Palas. All districts of the Hazara Division were included in the study for sampling.

Prior to sampling a questionnaire was designed to collect data about the milking buffaloes and farms from which milk samples were collected. Sampling was done using a three-stage sampling design, as previously described [12]. Eleven buffalo farms were chosen in each district of Hazara Division, randomly selected with the criteria that they be at least 5 km apart (all between 5–10 km) and contain at least 5 animals on the farm. On each farm five buffaloes were conveniently/randomly selected for sampling, for a total of 440 milking water buffaloes (any breed). Both symptomatic (clinical mastitis) and asymptomatic (sub-clinical or uninfected) animals were included in the random sampling. Buffaloes that were not milking, or those with fatal disease, were excluded from the study.

## Sampling and initial screening for bovine mastitis

Milk samples were collected from all four teats of each of the 440 buffaloes, with the length of each of the quarters (teat) measured (up to 1mm accuracy) using a ruler, and teat diameter at the apex, midpoint and base measured with a Vernier caliper. Udders were cleaned with 70% ethanol soaked cotton, then the first few drops of milk allowed to flow as waste, followed by the next few drops screened for bovine mastitis using the Surf Field Mastitis test (SFMT) with 4% surf solution (Excel, Uniliver, Pakistan). Positive samples, which formed clots and clumps when mixed with surf solution due to increased somatic cell count (SCC), were collected in a sterile falcon tube (15ml) and labelled. Negative milk samples (milk samples that did not form clots or clumps when mixed with 4% surf excel solution) were discarded immediately after testing. Positive samples were stored on ice until transported to the microbiology lab of the Department of Microbiology (University of Haripur) for further processing.

## Culturing of milk samples, *S. aureus* isolation and susceptibility testing

Milk samples were cultured on mannitol salt agar (MSA) plates, with 50 µl of each sample spread on the plate, and incubated for 24 hours at 37˚C. Staphylococci were identified according to standard microbiological procedures, with isolates that were mannitol fermenters, Gram-positive cocci (grape-like clusters), producing catalase, DNase and positive to tube coagulase tests (producing coagulase) with human plasma, considered *S. aureus* [13]. Further confirmation of *S. aureus* was done by polymerase chain reaction (PCR) detecting the *nuc* gene.

Antibiotic susceptibility was assessed using standard Clinical and Laboratory Standards Institute guidelines for antimicrobial disk diffusion using the following sixteen antibiotics: ampicillin (AM), amoxicillin (AX), lincomycin (L2), ceftazidime (CAZ), azithromycin (AZM), ceftriaxone (CRO), norfloxacin (NOR), cefoxitin (FOX), gentamicin (CN), erythromycin (E), tetracycline (TE), doxycycline (DO), clindamycin (DA), trimethoprim/sulfamethoxazole (SXT), rifampin (RA), and linezolid (LZN) (Oxoid, UK).

## Molecular characterization of the *S. aureus* isolates

DNA was extracted using the rapid boiling method [14]. A multiplex polymerase chain reaction (PCR) assay, capable of differentiating *S. aureus* from coagulase negative staphylococci (based on a *nuc* gene target), while also distinguishing MRSA from MSSA (based on a *mecA* gene target) and detecting the PVL gene [15], was used to confirm the earlier clinical strain assignation. *S. aureus* isolates were subsequently fingerprinted using pulsed field gel electrophoresis (PFGE) using a standardized protocol [16]. Briefly, *S. aureus* agarose plugs were digested with smaI and loaded onto a 1% agarose gel, then electrophoresis done with a CHEF mapper using switch times of 5.3 to 35 sec, for 18 hours at 14˚C, 6.0V/cm with an angle of 120 in 0.5x TBE. PFGE-generated DNA fingerprints were analyzed with BioNumerics Ver. 6.6 (Applied Maths, Sint-Martens-Lattem, Belgium) using a position tolerance of 1.5 as well as an optimization of 0 [16]. Strains were assessed for antiseptic resistance (*qacA/B*, *smr*) and antibiotic resistance genes (*mupA* and *mupB*) with a multiplex PCR assay [17]. Isolates were further characterized with SCC*mec* typing [18], accessory gene regulator (*agr*) typing [19], staphylococcal protein A (*spa*) typing [20], and multilocus sequence typing [21].

## Statistical analysis

The collected data was analyzed for the prevalence and association of risk factors with bovine mastitis in different districts of the Hazara Division, as well as for the prevalence of clinical and sub-clinical bovine mastitis. Binary logistic regression was used for two level variable

analysis, and multinomial logistic regression used for more than two levels. SPSS was used to determine Chi-Square values for the variables. A $p$ value of <0.05 was considered significant.

### Ethics approval

This study was approved by the ethics and research committee of the Department of Zoology, Hazara University, Mansehra, Pakistan (Reference No. hu.zool.rerc321; approval date: May 8, 2019).

## Results

### Overall epidemiology of *Staphylococcus* from Hazara division

A total of 440 milking water buffaloes were sampled in the study area, each of which has 4 independent quarters, for a total of 1760 quarters. Of the 440 animals sampled less than half (152; 34.55%) were found to be SFMT positive, indicating the presence of an infection. All the infections contained a bacterial pathogen, as the same percentage of samples (34.55%) resulted in organism growth. As shown in **Fig 1A**, nearly all of the animals with infections carried staphylococci, found in 140 (31.82%) of them, and *S. aureus* found in 81 (18.41%) of the animals. Twenty-seven (6.14%) animals were infected by both *S. aureus* and other coagulase negative staphylococci (CoNS). Only twelve (2.73%) animals were infected solely by non-staphylococcal bacteria.

Of the 1760 total quarters, 599 (34.03%) were infected and thus SFMT positive, with one fewer (598; 33.98%) resulting in bacterial growth. As seen in **Fig 1B**, animals were commonly infected by multiple bacterial species, with only 201 (11.42%) of the quarters carrying staphylococci, while 397 (22.56%) of the quarters were infected by other bacteria. *S. aureus* was isolated from 102 (5.80%) of the quarters, of which 16 (15.69%) were right front quarters (a), 15 (14.71%) were right rear quarters (b), 35 (34.31%) were left front quarters (c), whereas, 36 (35.29%) were left rear quarters (d), as shown in **Table 1**. This indicates that left quarters (69.6%) were more prone to infection than those on the right side (30.4%). Moreover, infection in a single quarter was the most common occurrence, representing 75.31% of the infections, with only 2 animals (2.46%) infected in 3 or more quarters (**Table 1**).

Next we looked at the epidemiology of clinical vs sub-clinical cases of bovine mastitis at the farm, animal and quarter level, and noted that at all levels sub-clinical cases were more common. Of the 440 animals tested, 61 (75.31%) presented with sub-clinical infection, while 20 (24.69%) presented with clinical infection, as summarized in **Table 2**.

Finally, the data was broken down into a district-wise analysis, which indicated that there was a widespread and even distribution of infection across the districts, with the exception of Kolai Palas which had fewer cases. Looking specifically at the 102 *S. aureus* infected animals, we saw an even distribution of cases across the districts, as summarized in **Table 3** and **Fig 2**. Among the 599 infected quarters, 96 (16.02%) were from animals in Haripur, 69 (11.51%) from Abbottabad, 76 (12.69%) from Mansehra, 72 (12.02%) from Battagram, 89 (14.86%) from Torghar, 92 (15.35%) from Kohistan Upper, 68 (11.35%) from Kohistan Lower, and 37 (6.17%) from Kolai Palas.

### Association between *S. aureus* infection and the various risk factors associated with mastitis in buffaloes

Various risk factors for bovine mastitis in buffaloes, including breed, age, parity, shelter, milk color, duration of lactation, udder shape, quarter length, quarter shape, quarter lesion, udder condition and treatment, were analyzed to determine if they were associated with infection. As

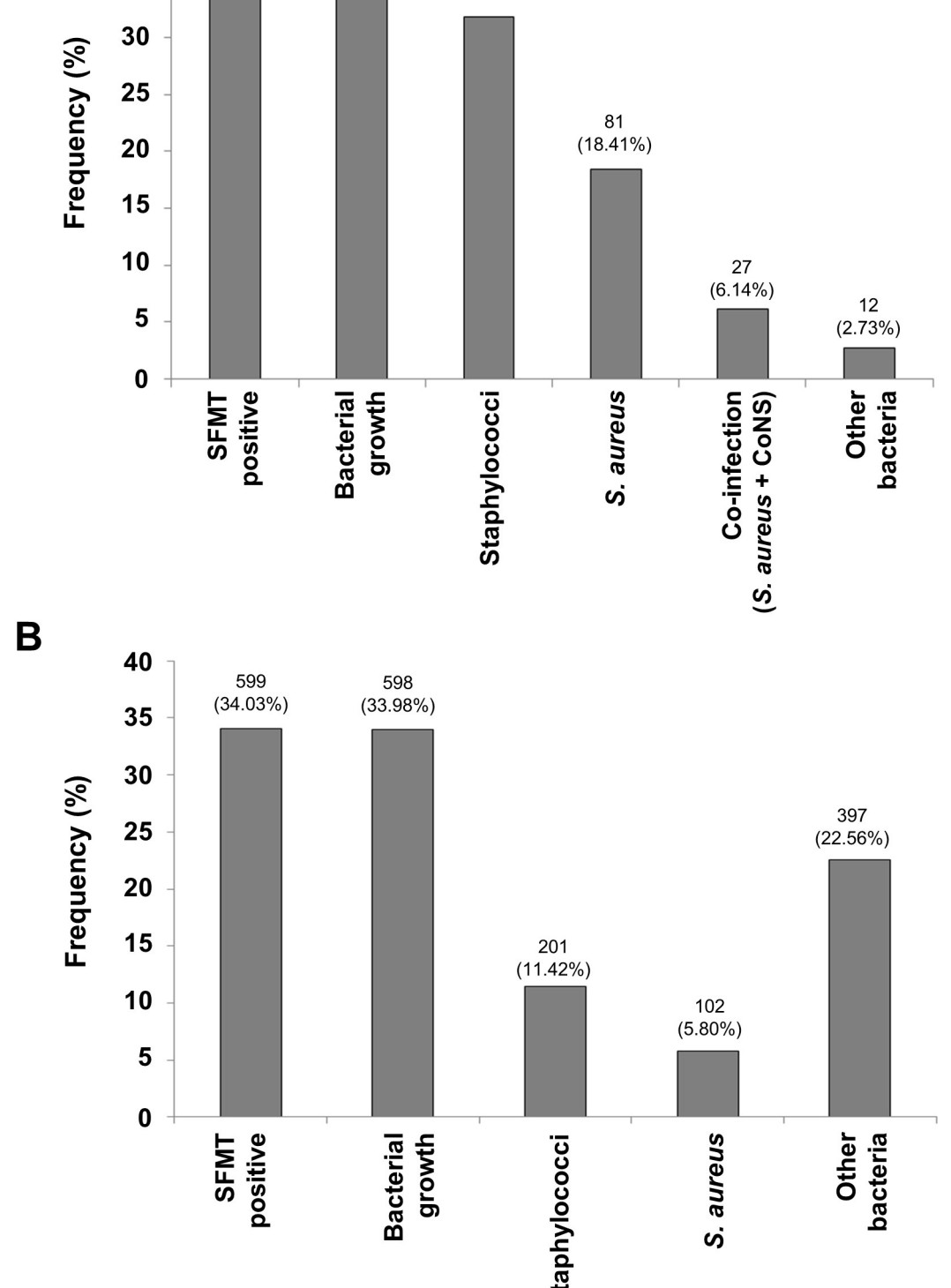

**Fig 1.** Frequency of bovine mastitis infection at the animal (A) and quarters (B) levels. The number of samples that were SFMT positive, resulted in bacterial growth, contained staphylococci, contained *S. aureus*, contained *S. aureus* with

coagulase negative staphylococci (CoNS), and only contained other bacteria are indicated above the curve, with corresponding percentages in brackets.

summarized in **Table 4**, breed ($p = 0.865$), age ($p = 0.58$), parity ($p = 0.923$), duration of lactation ($p = 0.936$), and udder shape ($p = 0.553$) did not show a statistical association with disease. Shelter, however, was linked to disease ($p < 0.05$), as was milk color ($p < 0.05$) which, with an odds ratio of 33.09 (95% CI, 22.01–35.45), indicated that yellow milk was more linked to mastitis. Quarter length was also associated with disease ($p < 0.05$), as was teat/quarter shape ($p < 0.05$), with cylindrical (OR, 14.3; 95% CI, 14.30–15.65) and round (OR, 20.58; 95% CI, 2.37–178.75) shapes having an increased link to infection. Similarly, udder lesion (88.01, $p < 0.05$), specifically the presence of a laceration (OR, 30.67; 95% CI, 24.25–35.75), is another variable with significantly increased association with infection. Udder condition ($p < 0.05$) was also linked to disease, with the presence of inflammation (OR, 33.59; 95% CI, 1.88–35.11) having a higher odds of disease. Antibiotic treatment is a final variable associated with disease ($p < 0.05$), with no treatment having an increased link to mastitis infection (OR, 9.45; CI, 7.88–11.65).

## Molecular epidemiology of *S. aureus* isolated from buffaloes in Hazara division

The 102 *S. aureus* isolates were subjected to molecular analysis, with results summarized in **Table 5**, and the full results in S1 **Table**. Methicillin-resistance levels were low in the region, with 82 (80.4%) of the isolates methicillin-sensitive (MSSA), while only 20 (19.6%) were methicillin-resistant (MRSA). Within the MSSA three *spa* types were identified, including t2078 (n = 12, 14.6% of the MSSA), t7286 (n = 25, 30.5% of the MSSA), and t7867 (n = 45, 54.9% of the MSSA). The MSSA were all grouped into two MLST and *agr* types, including ST101-*agrI* (n = 12, 14.6% of the MSSA) and ST9-*agrII* (n = 70, 85.4% of the MSSA). In contrast, all of the MRSA shared a single *spa* (t8834), MLST (ST22) and *agr* (I) type, and all carried SCC*mec* type IVa. None of the isolates carried the PVL genes or ACME cassette.

Based on the PFGE analysis four main strain types were identified in the region, with minor variations of PFGE banding pattern observed within each type (**Fig 3**). Twelve isolates belonged to ST9-t7867-MSSA (*agrII*), sharing 59% or greater similarity in Dice coefficient of correlation (DCC). Fifty-eight isolates belonged to the second group, which shared the same MLST (ST9) as the first group, but occupied to a distinct arm of the dendrogram. This group of ST9-MSSA (*agrII*) all shared 61% or greater similarity in DCC, but only 50% similarity with

**Table 1. Summary of the *S. aureus* positive bovine mastitis infected quarters of buffaloes in Hazara division.**

|  | *S. aureus* positive | | | |
|---|---|---|---|---|
|  | **No.** | **%** | **$X^2$** | **$p$-value** |
| Right Front Quarter (a) | 16 | 15.69 | 16.69 | 0.000817 |
| Right Rear Quarter (b) | 15 | 14.71 | | |
| Left Front Quarter (c) | 35 | 34.31 | | |
| Left Rear Quarter (d) | 36 | 35.29 | | |
| 1 infected quarter | 61 | 75.31 | 122.95 | <0.00001 |
| 2 infected quarters | 19 | 23.46 | | |
| 3 infected quarters | 1 | 1.23 | | |
| 4 infected quarters | 1 | 1.23 | | |

Note: No., number of animals; %, percentage.

**Table 2. Details of clinical and sub-clinical bovine mastitis cases at the farm, animal and quarter level.**

| Level | Overall | Overall Mastitis | | Clinical Mastitis | | Sub-Clinical Mastitis | |
|---|---|---|---|---|---|---|---|
| | No. | No. | % | No. | % | No. | % |
| Farm | 88 | 62 | 70.45 | 20 | 32.26 | 52 | 83.87 |
| Animal | 440 | 81 | 18.41 | 20 | 24.69 | 61 | 75.31 |
| Quarter | 1760 | 102 | 5.80 | 32 | 31.37 | 70 | 68.63 |

Note: No., number; %, percentage.

the previously first group. Two *spa* types, t7286 (n = 25) and t7867 (n = 32), were intermixed in this group. The third strain type was ST101-t2078-MSSA (*agrI*), accounting for 12 of the isolates, all sharing 82% or greater similarity in DCC. The final strain type was ST22-t8934-MRSA-IVa (*agrI*), representing all 20 of the MRSA, sharing 67% similarity in DCC.

In general each strain type was isolated from several districts, and multiple farms in each district, as shown in **Figs 2** and **3**. ST9-t7867-MSSA was isolated from 4 of the districts, including Abbottabad, Haripur, Upper Kohistan, and Torghar. The closely related ST9-MSSA was isolated from all 8 districts, while ST101-t2078-MSSA was isolated from 5 districts, including Abbottabad, Upper Kohistan, Torghar, Mansehra and Battagram. Finally, ST22-t8934-MRSA-IVa was isolated from all the districts except upper Kohistan. It is worth noting that 14 of the 17 isolates (representing 12 of 14 animals) from Upper Kohistan belonged to strain type ST9-MSSA, as did 7 of the 10 isolates (representing 5 of 8 animals) from Lower Kohistan.

Interestingly, there were ten instances where different strain types were isolated from a single animal, with each strain isolated from different quarters within the animal, and all occuring on different farms. Two animals carried both ST9-t7867-MSSA and ST101-t2078-MSSA, two animals carried both ST9-t7867-MSSA and ST22-t8934-MRSA-IVa, two animals carried both ST9-t7867-MSSA and ST9-MSSA, two animals carried both ST101-t2078-MSSA and ST9-MSSA, and one animal carried both ST22-t8934-MRSA-IVa and ST9-MSSA. One animal also carried two ST9-MSSA belonging to the same PFGE group, but that had different *spa* types (t7867 and t7286).

The presence of antiseptic and antibiotic resistance genes was also assessed by PCR amplification. None of the isolates were positive for the multidrug efflux pump genes (*qacA*, *qacB*), small multidrug resistance gene (*smr*), or mupirocin resistance genes (*mupA* or *mupB*).

**Table 3. District-wise details of the bovine mastitis clinical and sub-clinical cases in *S. aureus* infected animals.**

| | | District | | | | | | | | | | | | | | |
|---|---|---|---|---|---|---|---|---|---|---|---|---|---|---|---|---|
| Level | Mastitis | Haripur | | Abbottabad | | Mansehra | | Battagram | | Torghar | | Kohistan Upper | | Kohistan Lower | | Kolai Palas | |
| | | No. | % | No. | % | No. | % | No. | % | No. | % | No. | % | No. | % | No. | % |
| Animal | Overall | 12 | 14.81 | 10 | 12.35 | 13 | 16.05 | 10 | 12.35 | 12 | 14.81 | 14 | 17.28 | 8 | 9.88 | 2 | 2.47 |
| | clinical | 4 | 20 | 3 | 15 | 2 | 10 | 2 | 10 | 3 | 15 | 3 | 15 | 2 | 10 | 1 | 5 |
| | sub-clinical | 8 | 13.11 | 7 | 11.8 | 11 | 18.03 | 8 | 13.11 | 9 | 14.75 | 11 | 18.03 | 6 | 9.84 | 1 | 1.64 |
| Quarter | Overall | 17 | 16.67 | 15 | 14.71 | 15 | 14.71 | 12 | 11.76 | 14 | 13.73 | 17 | 16.67 | 10 | 9.80 | 2 | 1.96 |
| | clinical | 8 | 25 | 6 | 18.76 | 3 | 9.38 | 3 | 9.38 | 4 | 12.5 | 5 | 15.63 | 2 | 6.25 | 1 | 3.13 |
| | sub-clinical | 9 | 12.86 | 9 | 12.86 | 12 | 17.14 | 9 | 12.86 | 10 | 14.29 | 12 | 17.14 | 8 | 11.43 | 1 | 1.43 |

Note: No., number; %, percentage.

### Antibiotic resistance phenotypes of the *S. aureus* isolates

Resistance to 16 different antibiotics was assessed in the 102 *S. aureus* isolates. In general the strains showed low levels of resistance to the antibiotics tested, as shown in **Table 6**. Notable exceptions were ceftazidime, with 100% of the isolates showing resistance, and ampicillin, with 42% of the isolates showing resistance. Approximately 40% of the isolates showed intermediate resistance to cefoxitin and lincomycin, and approximately 57% to ceftriaxone.

Greater resistance was seen among the ST22-MRSA than amongst the MSSA, with a higher proportion of MRSA resistant to ampicillin (MSSA-29%, MRSA-90%), cefoxitin (MSSA-0%, MRSA-100%), amoxicillin (MSSA-23%, MRSA-55%), doxycycline (MSSA-0%, MRSA-20%), trimethoprim-/sulfamethoxazole (MSSA-8.5%, MRSA-30%), linezolid (MSSA-1.2%, MRSA-10%), azithromycin (MSSA-1.2%, MRSA-25%), ceftriaxone (MSSA-1.2%, MRSA-30%), tetracycline (MSSA-1.2%, MRSA-20%), and erythromycin (MSSA-1.2%, MRSA-10%). There wasn't a link between antibiotic resistance and the MSSA/MRSA strain type, region or farm from which the samples came (details available in **S1 Table**).

## Discussion

Bovine mastitis remains the most prevalent and economically consequential diseases of dairy animals, and *Staphylococcus aureus* is the most common causative agent. In the current study

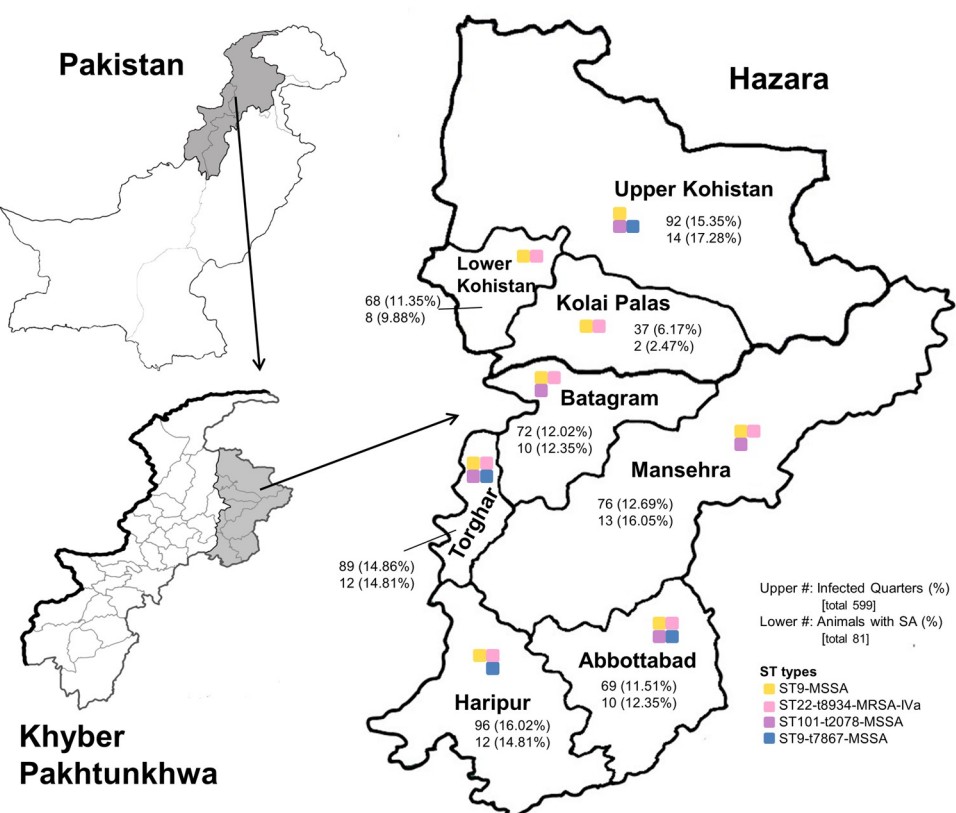

**Fig 2. Frequency of bovine mastitis, and presence of the *S. aureus* strain types in each district of the Hazara division.** The Khyber Pakhtunkhwa province is shown shaded grey in the Pakistan map, as is the Hazara division in the KP map. Individual districts in the Hazara division are shown. The upper number in each district is the number of infected quarters (percentage of the 599 total infected quarters), while the lower number is the number of animals infected with *S. aureus* (percentage of the 81 total *S. aureus* infected animals). Coloured squares represent the presence of a strain type in the district, with yellow for ST9-MSSA, pink for ST22-t8934-MRSA-IVa, purple for ST101-t2078-MSSA, and blue for ST9-t7867-MSSA. #, number; %, percentage; SA, *S. aureus*.

**Table 4. Analysis of the association between animal-level bovine mastitis and its various risk factors.**

| Variables | Buffaloes | | 95% CI | χ2 (p-value) |
|---|---|---|---|---|
| | No. Infected/ total | % Infected | OR | |
| **Breed** | | | | |
| Nili Ravi | 59/311 | 18.97 | 1.44 (0.90–5.01) | 0.29 (0.865) |
| Khundi | 19/109 | 17.43 | 1.95 (0.55–6.84) | |
| Azikheli | 3/20 | 15 | 2.28 (0.62–8.34) | |
| **Age (Years)** | | | | |
| 2 | 3/9 | 33.33 | 4.92 (0.52–47.07) | 2.85 (0.58) |
| 3 | 6/44 | 13.64 | 1.64 (0.55–4.94) | |
| 4 | 44/255 | 17.25 | 1.84 (0.73–4.63) | |
| 5 | 26/111 | 23.42 | 1.74 (0.66–4.62) | |
| 6 | 2/21 | 9.52 | 1.10 (0.45–4.02) | |
| **Parity** | | | | |
| 1 time | 9/47 | 19.15 | 0.86 (0.26–2.83) | 0.48 (0.923) |
| 2 times | 54/302 | 17.88 | 0.86 (0.31–2.41) | |
| 3 times | 13/69 | 18.84 | 0.72 (0.23–2.22) | |
| 4 times | 5/22 | 22.73 | 1.19 (0.56–3.25) | |
| **Shelter** | | | | |
| Barn | 66/355 | 18.59 | 0.38 (0.25–0.40) | 416.44 (<0.05)* |
| House | 15/85 | 17.65 | 0.32 (0.16–0.65) | |
| **Milk colour** | | | | |
| White | 64/412 | 15.53 | 1.70 (0.99–3.25) | 481.29 (<0.05)* |
| Yellow | 17/28 | 60.71 | 33.09 (22.01–35.45) | |
| **Duration of lactation (months)** | | | | |
| >0.9–3 | 4/20 | 20 | 0.78 (0.15–3.93) | 0.13 (0.936) |
| >3–6 | 4/25 | 16 | 1.09 (0.33–3.57) | |
| >6 | 73/395 | 18.48 | 1.40 (0.44–4.23) | |
| **Udder shape** | | | | |
| Round | 44/248 | 17.74 | 2.55 (0.95–3.45) | 1.18 (0.553) |
| Cup | 25/116 | 21.55 | 0.38 (0.17–0.88) | |
| Bowl | 12/76 | 15.79 | 0.97 (0.55–1.71) | |
| **Right front quarter length** | | | | |
| 5.0–5.5 cm | 19/58 | 32.76 | 4.04 (0.70–23.15)[†] | 11.56 (<0.05)* |
| 5.6–6.0 cm | 37/192 | 19.27 | | |
| 6.1–6.5 cm | 24/181 | 13.26 | | |
| 6.6–7.0 cm | 1/9 | 11.11 | | |
| **Right rear quarter length** | | | | |
| 5.0–5.5 cm | 2/2 | 100 | 0.74 (0.29–1.88)[†] | 12.96 (<0.05)* |
| 5.6–6.0 cm | 50/232 | 21.55 | | |
| 6.1–6.5 cm | 16/114 | 14.04 | | |
| 6.6–7.0 cm | 13/92 | 14.13 | | |
| **Left front quarter length** | | | | |
| 5.0–5.5 cm | 10/31 | 32.26 | 90.18 (0.03–1.15)[†] | 9.44 (<0.05)* |
| 5.6–6.0 cm | 48/234 | 20.51 | | |
| 6.1–6.5 cm | 16/141 | 11.35 | | |
| 6.6–7.0 cm | 7/34 | 20.59 | | |
| **Left rear quarter length** | | | | |
| 5.0–5.5 cm | 7/28 | 25 | 1.034 (0.32–3.36)[†] | 50.86 (<0.05)* |
| 5.6–6.0 cm | 16/21 | 76.19 | | |
| 6.1–6.5 cm | 45/299 | 15.05 | | |
| 6.6–7.0 cm | 13/92 | 14.13 | | |
| **Teat/quarter shape** | | | | |
| Cylindrical | 59/394 | 14.97 | 14.30 (14.30–15.65) | 36.39 (<0.05)* |
| Round | 9/24 | 37.5 | 20.58 (2.37–178.75) | |
| Pointed | 8/16 | 50 | 1.67 (0.16–17.26) | |
| Flat | 5/6 | 83.33 | 0.71 (0.05–9.70) | |

(*Continued*)

**Table 4.** (Continued)

| Variables | Buffaloes | | 95% CI | χ2 (p-value) |
|---|---|---|---|---|
| | No. Infected/ total | % Infected | OR | |
| **Teat/quarter lesion** | | | | |
| None | 62/421 | 14.73 | 2.34 (1.25–4.26) | 88.01 (<0.05)* |
| Laceration | 15/15 | 100 | 30.67 (24.25–35.75) | |
| Edema | 4/4 | 100 | 1.45 (0.27–2.85) | |
| **Udder condition** | | | | |
| Normal | 62/410 | 15.12 | 1.73 (0.87–3.65) | 60.79 (<0.05)* |
| Inflammation | 4/12 | 33.33 | 33.59 (1.88–35.11) | |
| Haemorrhage | 11/11 | 100 | 1.00 (0.25–1.26) | |
| Necrosis | 2/3 | 66.67 | 1.00 (0.25–1.26) | |
| Cord formation | 2/4 | 50 | 1.00 (0.25–1.26) | |
| **Treatment** | | | | |
| None | 47/326 | 14.42 | 9.45 (7.88–11.65) | 88.10 (<0.05)* |
| Prophylactic | 15/95 | 15.79 | 2.21 (1.74–3.98) | |
| Post infection | 19/19 | 100 | 1.04 (0.99–1.50) | |

Note: No, Number; %, Percentage; OR, Odd ratio; CI, Confidence Interval. Binary logistic regression was used for two levels and multinomial logistic regression used for more than two levels variables.; χ2 (p-value), Chi-Square value (*Pearson values* for the variable groups as a whole); *, Significantly associated with the disease (*p-value* <0.05)

†, The variable is continuous therefore the values were calculate for the whole variable.

we characterized staphylococci in lactating buffaloes from the eight districts of Hazara Division, KP, Pakistan. Our study represents the most comprehensive characterization of staphylococcal isolates from cases of bovine mastitis in the region.

From an epidemiological standpoint, results from our study were in keeping with previously reported studies on bovine mastitis in Pakistan. We identified mastitis in 34.55% of the animals tested, within the range of prevalence rates previously described in the country. In the Punjab region rates of 44% and 31.75% were found, while in the Faisalabad District rates ranged from 19.74–25.12%, and in the Peshawar District mastitis was found in 36.35% of buffaloes in rural regions [8,9,22,23]. Looking specifically at staphylococci, we isolated the genus from 31.8% of the animals, with *S. aureus* isolated from 18.41% of the total animals, and 5.8% of the samples (quarters). Our *S. aureus* prevalence does fall below rates from previous studies where approximately 27% of animals, or 28–50% of samples contained *S. aureus* [8,9,24,25]. Differential husbandry practices and/or disease management systems between the farms likely

**Table 5. Overall molecular characteristics of the *S. aureus* isolated from bovine mastitis cases in the Hazara division.**

| Isolate No. (%) | Type No. (%) | No. (%) | Molecular Characterization | | | | Antiseptic*/Antibiotic† Resistance Genes | | | |
|---|---|---|---|---|---|---|---|---|---|---|
| | | | ST Type | *spa* Type | *agr* Type | *SCCmec* Type | qacA/B* | smr* | mupA† | mupB† |
| *S. aureus* 102 (50.7%) | MSSA 82 (80.4%) | 12 (14.6%) | ST101 | t2078 | I | - | 0 (0%) | 0 (0%) | 0 (0%) | 0 (0%) |
| | | 25 (30.5%) | ST9 | t7286 | II | - | | | | |
| | | 45 (54.9) | ST9 | t7867 | II | - | | | | |
| | | | | | | | | | | |
| | MRSA 20 (19.6%) | 20 (100%) | ST22 | t8934 | I | IVa | | | | |

Note: No., number; %, percentage; SCC*mec*, staphylococcal cassette chromosome *mec*; *agr*, accessory gene regulator; *spa*, staphylococcal protein A; ST, Staphylococcal type; qacA/B, multidrug efflux pump; smr, multidrug resistance protein family; mupA/B, mupirocin resistance A/B genes; -, negative

*, representing antiseptic resistance genes

†, representing antibiotic resistance genes.

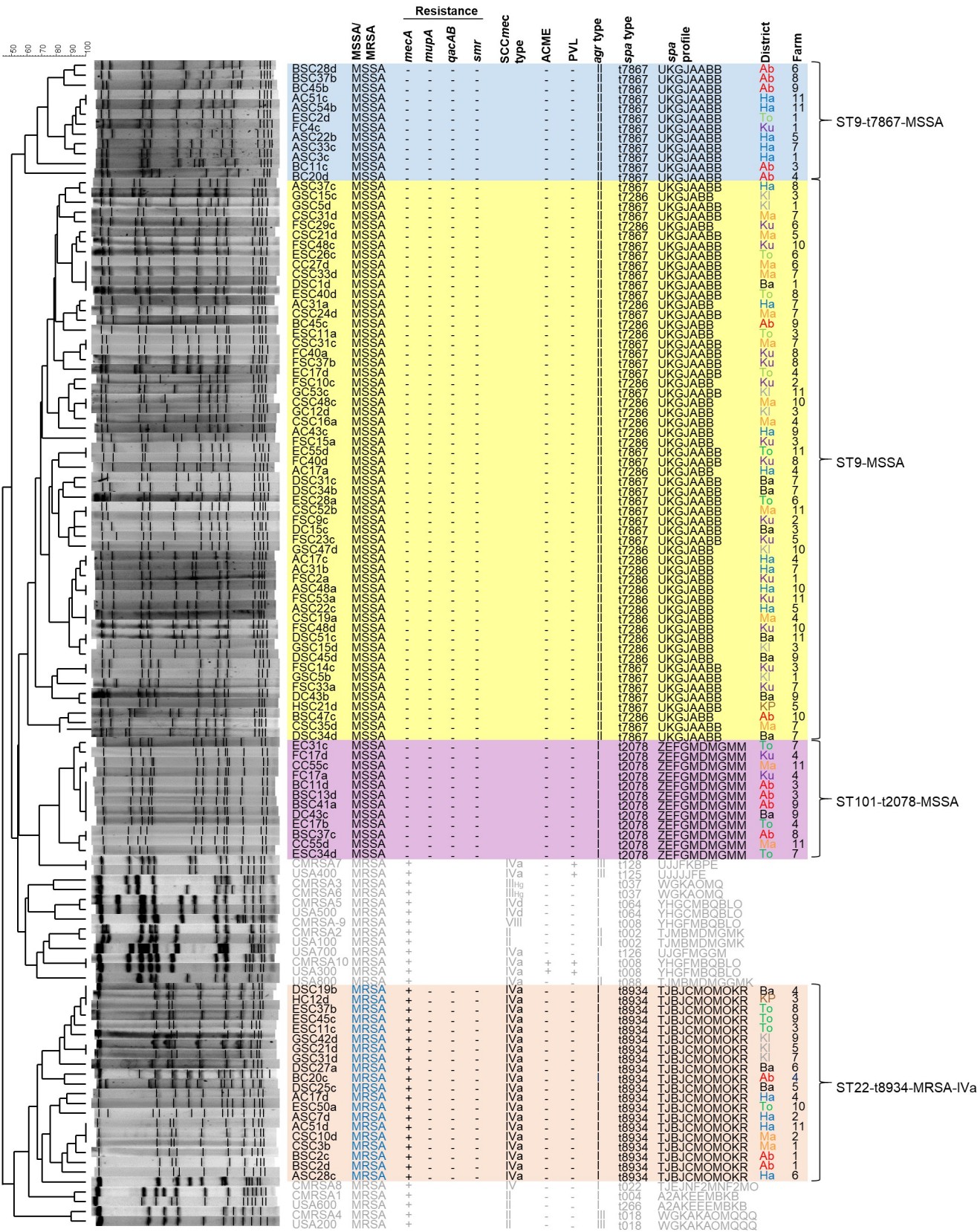

**Fig 3. Pulsed field gel electrophoresis fingerprints of all *S. aureus* isolates from the Hazara district of Pakistan.** Molecular characteristics and corresponding district and farm information are shown. ST9-t7867-MSSA are shaded blue, ST9-MSSA are shaded yellow, ST101-t2078-MSSA are shaded purple, and ST22-t8934-MRSA-IVa are shaded pink. Reference strains types from Canada (CMRSA-) and USA (USA-) are indicated in grey font. *mecA*, methicillin resistance gene A; *mupA*, mupirocin resistance gene A; *qacAB* and *smr*, efflux mediated antiseptic resistance genes; SCC*mec*, staphylococcal cassette chromosome *mec*; ACME, arginine catabolic mobile element; PVL, Panton Valentine Leukocidin; *agr*, accessory gene regulator; *spa*, staphylococcal protein A; -, negative for gene; + positive for gene; Ha, Haripur (blue font); Ab, Abbottabad (red font); Ma, Mansehra (orange font); Ba, Battagram (black font); Ku, Kohistan Upper (purple font); Kl, Kohistan Lower (grey font); To, Torghar (green font); KP, Kolai Palas (brown font).

contribute to any variations noted. The farms in this study were all private, operating on a very small commercial scale. Cleanliness was being maintained on the farms and infection control practices were followed, however, there were variations in terms of the timing of antibiotic treatment and how strictly isolation and treatment of infected animals was followed. Pakistan is surrounded by tropical and subtropical environments, both of which support the growth of organisms responsible for mastitis [10]. Pathogens can be transmitted by dirty hands, contaminated equipment, soil and insects, therefore differential mitigation of these environmental conditions can adversely affect animal health, and consequently the rates of mastitis seen in the different studies.

Also noted in this study was the fact that sub-clinical mastitis infection (75.31%) was more prevalent than was clinical mastitis (24.69%). This is similar to previously reported results such as in the Pothohar region of Punjab, where subclinical mastitis was identified in 67.3% of buffaloes [11], and in the Lahore district where it was identified in 59.64% of buffaloes [23]. Sub-clinical mastitis is a significant concern as no overt symptoms of disease are seen, therefore it can persist undetected for long periods in a herd and spread rapidly between animals. Clinical mastitis, on the other hand, rapidly leads to udder redness, changes in milk condition and sticky discharge, permitting speedy identification and treatment of an infected animal. The consistently high rates of sub-clinical mastitis seen in this and other studies emphasize the importance of adherence to prescribed infection control practices, as well as the need for vigilance and early detection methods. Regular screening for elevated somatic cell count (SCC)

**Table 6. Antibiotic resistance of the *S. aureus* from Hazara division.**

| Antibiotic | Percentage Resistant | Percentage Intermediate Resistance | Percentage Sensitive |
|---|---|---|---|
| Ampicillin | 42.16 | 0 | 57.84 |
| Cefoxitin | 19.61 | 38.24 | 42.16 |
| Clindamycin | 0 | 12.75 | 87.25 |
| Gentamycin | 1.96 | 0 | 98.04 |
| Amoxicillin | 29.41 | 0 | 70.59 |
| Doxycycline | 2.94 | 0 | 97.06 |
| Lincomycin | 26.47 | 39.22 | 34.31 |
| Ceftazidime | 100 | 0 | 0 |
| Rifampin | 2.94 | 3.92 | 93.14 |
| TMP-SMX | 12.75 | 5.88 | 81.37 |
| Linezolid | 2.94 | 0.00 | 97.06 |
| Azithromycin | 7.84 | 10.79 | 81.37 |
| Ceftriaxone | 6.86 | 56.87 | 36.27 |
| Tetracycline | 5.88 | 3.92 | 90.2 |
| Norfloxacin | 0 | 1.96 | 98.04 |
| Erythromycin | 2.94 | 18.63 | 78.43 |

Note: TMP-SMX, Trimethoprim-sulfamethoxazole.

would offer the ideal detection system, however it would not be practical for routine use on the farm, therefore alternative screening methods are needed. In our study we noted that factors such as milk color, quarter length, quarter shape and lesion presence, udder condition and antibiotic treatment were all associated with disease. While we saw no association with breed, age, parity, shelter, duration of lactation and udder shape, they too have been associated with disease in other studies [1,10,11,22,26]. Given that regular screening of SCC is not possible on a farm, monitoring for these other associated factors would provide an alternative way to quickly identify sub-clinical infection, and curb its spread.

Looking at infections at the quarters level, previous studies have indicated that in buffaloes rear quarters are more prone to infection than front quarter [9,22,23], however, that was not true in our study. We observed that left quarters (both front and rear) had more than twice the infection rate (69.6%) of right quarters (30.4%), which held true in all districts and on all farms. While the rate of infection in the left rear quarter was higher than that in the left front quarter, the left front quarter still had a higher infection rather than the right rear quarter. The reasons why our results differ from previous reports remains unclear, but it likely comes back to milking and infection control practices on the farms. Buffaloes on the farms in this study were all hand milked and it has been reported that in Pakistan milking is primarily done from the left side, meaning the left teats are often being touched while reaching for the right quarters, increasing their chances of contamination [27]. As such, increased attention may need to be paid to sterilization of teats on the milking side in hand milked animals. Cleanliness of the farm would also play a role, as contamination of rear quarters is influenced by their increased exposure to dirt when animals lay on the ground, as well as their increased exposure to urine and fecal material. With cleanliness being well maintained on these smaller farms, the rear quarters would be at a lower risk of infection. The tendency of buffaloes to lay on their left side would also increase exposure of left quarters to injury and infection, increasing their chances of infection.

From a molecular standpoint this study presents detailed and important information, since genotyping of *S. aureus* from Pakistan is not widely done. A thorough understanding of strains affecting both animals and humans in Pakistan is needed to determine if there is widespread transfer from animals to humans, or vice versa, and how it impacts the overall epidemiology of *S. aureus* in the country. We were unable to find comparable molecular data for strains causing buffalo mastitis, however, in one study that characterized MRSA from table eggs in Haripur two PFGE patterns, representing ST8-t8645-MRSA-IV, ST772-t657-MRSA-IV and ST772-t8645-MRSA-IV were identified [28]. In a study characterizing *S. aureus* from mastitis in dairy cows in China, isolates belonging to a wide range of clonal complexes (CCs) (15 different CC, encompassing 19 different sequence types) were identified [29]. Several *spa* types were noted for most of the ST types, and most isolates (93.4%) were methicillin-sensitive. *S. aureus* from a large variety of CCs were likewise isolated from cow milk samples in other parts of the world, however it is impossible to draw meaningful comparisons as they are in different species or geographical locations [30–34]. In our study 80.4% of the isolates were methicillin sensitive, but differed dramatically from the other studies in that only three ST types were identified. Based on PFGE analysis these were divided into four strain types, including ST9-t7867-MSSA, ST9-MSSA (which clustered apart from the other ST9 in the PFGE dendrogram), ST101-t2078-MSSA and ST22-t8934-MRSA-IVa. This suggests that there is very limited diversity in the *S. aureus* circulating amongst buffalo farms in this region of Pakistan, which could be due to the close geographic connection between farms and districts in the division, as well as their homogenous climate. With similar environmental factors (such as humidity and temperature) driving selection, similar strain types might prove to be the most fit in each district of this region. Additionally, the farms are small scale and just 5 km apart, meaning

the same people could be visiting multiple farms to purchase milk, acting as sources of pathogen transfer. On top of that, the owners of farms in close proximity are often relatives and there can be an exchange of animals between farms, again being a source of pathogen transfer. Comparing the animal associated strains to those in human infections, we observed that our MSSA were different from the human-associated *S. aureus* identified in Pakistan, while the MRSA were similar to those identified in one investigation. *S. aureus* in humans (particularly in tertiary care facilities), were primarily identified as CC121-MSSA, ST772-MRSA-V, CC8-MRSA and ST239-MRSA [35–39]. In one study CC22-MRSA-IV was identified in 13.6% (belonging to three different strains) of the human isolates [35], matching the ST type of all the MRSA in our study. As such, it appears that human-associated *S. aureus* in Pakistan are also limited in diversity and, with one exception, different from the buffalo-associated lineages we identified. Caution must, however, be taken with this conclusion since there are only a few studies available for the country, all with limited sample size, and none within the same geographic location as our study. As well, only one of the studies characterized MSSA, leaving open the possibility that our animal-associated MSSA could yet be identified in humans when a larger number of isolates are tested.

In terms of the strains we identified, 68.6% of the isolates belonged to strain type ST9, all being MSSA. ST9 is a predominant livestock-associated *S. aureus*, particularly in Asian countries like China [40,41] Taiwan [42], and Malaysia [43]. This strain type is frequently detected in pig farms and food products of animal origin, as well found colonizing both animals and human hosts [44], and causing infections in humans [45,46]. Interestingly, ST2454-t7867, a single locus variant of our ST9-t7867, is believed to be commonly associated with bovine mastitis in nearby India [47]. The remainder of the MSSA in this study belonged to strain type ST101, which is encountered in many places across the globe, colonizing and infecting both animals and humans [48–52]. ST101 has also been associated with bulk milk samples and mastitis in cattle [53–55]. All MRSA identified in this study belonged to sequence type ST22, related to the epidemic strain EMRSA-15. This strain is prevalent in both animals (recognized as a causative agent in bovine mastitis) and humans, particularly in Eastern Asia and nearby India where it is believed to be replacing the ST239 clone in hospitals [56–62]. While EMRSA-15 and related ST22 are typically PVL positive, all of our mastitis-associated ST22-MRSA differed in that they were PVL negative, as has been noted for ST22 from tertiary care hospitals in the country [35].

On a final note, our study indicated that there was, in general, a relatively low level of antibiotic resistance amongst the *S. aureus* isolates. Exceptions were higher levels of resistance to penicillins (ampicillin-42%, amoxacillin 29%), and lincomycin (26.47%). Antibiotic use in food-producing animals is unregulated in Pakistan, with the actual usage unknown. A few studies have looked at commercial farms and determined that antibiotic use in cattle is above international averages, with beta-lactams, aminoglycosides and tetracyclines amongst the classes most commonly chosen [63,64]. Aminopenicillins, in particular, are commonly used for the treatment and prevention of bovine subclinical mastitis, with as high as 36.5% of market milk samples containing beta-lactam residues above the permissible level [65,66]. Unfortunately this study is limited in that the specific antibiotic use for each farm was not documented, therefore cannot be tied directly to the isolates, however future studies could investigate this.

## Conclusion

*S. aureus* remains the primary pathogen of concern in cases of bovine mastitis. This study presents the most complete molecular typing data for both methicillin-sensitive and methicillin-resistant *S. aureus* causing bovine mastitis in the Hazara district of Pakistan, and the country

as a whole. With limited molecular data currently available, our study provides crucial background information for future researchers, and could help elucidate the epidemiological spread of a pathogen that causes significant economic impact on a major industry in the country.

## Supporting information

**S1 Table. Summary of the *Staphylococcus aureus* isolated from buffaloes in Hazara division.**
(PDF)

## Author Contributions

**Conceptualization:** Muhammad Ali Syed, Kunyan Zhang.

**Funding acquisition:** Kunyan Zhang.

**Investigation:** Salma Javed, JoAnn McClure, Osahon Obasuyi, Shahzad Ali, Sadia Tabassum, Mohammad Ejaz.

**Resources:** Muhammad Ali Syed.

**Supervision:** Muhammad Ali Syed, Kunyan Zhang.

**Writing – original draft:** Salma Javed, JoAnn McClure.

**Writing – review & editing:** Muhammad Ali Syed, Kunyan Zhang.

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
