## [Decision Letter · Decision Letter 0]

1 Mar 2022

PONE-D-22-02764Epidemiology and Molecular Characterization of Staphylococcus aureus Causing Bovine Mastitis in Water Buffaloes from the Hazara Division of Khyber Pakhtunkhwa, PakistanPLOS ONE

Dear Dr. Zhang,

Thank you for submitting your manuscript to PLOS ONE. After careful consideration, we feel that it has merit but does not fully meet PLOS ONE’s publication criteria as it currently stands. Therefore, we invite you to submit a revised version of the manuscript that addresses all the points raised during the review process.

We look forward to receiving your revised manuscript.

Kind regards,

Herminia de Lencastre, Ph.D.

Academic Editor

PLOS ONE

Journal Requirements:

Reviewers' comments:

Reviewer's Responses to Questions

**Comments to the Author**

1. Is the manuscript technically sound, and do the data support the conclusions?

Reviewer #1: Partly

Reviewer #2: Yes

2. Has the statistical analysis been performed appropriately and rigorously? 

Reviewer #1: Yes

Reviewer #2: Yes

3. Have the authors made all data underlying the findings in their manuscript fully available?

Reviewer #1: Yes

Reviewer #2: Yes

4. Is the manuscript presented in an intelligible fashion and written in standard English?

Reviewer #1: Yes

Reviewer #2: Yes

5. Review Comments to the Author

Reviewer #1: Javed et al described the epidemiology and the prevalence of staphylococci causing bovine mastitis in water buffaloes in Hazara division of Pakistan. The authors used PFGE to assess the genetic relatedness of the S. aureus isolates recovered in the study and assessed them for antibiotic resistance. The study is well designed and should add to the growing data and knowledge of MRSA in food production animals in the region

Below are some comments for consideration

Minor comments

CoNS was not defined anywhere in the manuscript

Except when starting a sentence, “staphylococci” or “staphylococcal” should not be written as “Staphylococci”

Authors should consider reviewing the punctuations the manuscript.

Major comments

The introduction is well written and very easy to read. It gave a detailed background of the study and identified the knowledge gaps this study aimed to fill. I would not want to assume that there is no single study from Pakistan that report the molecular characterization either from human, animals, or environmental matrices. Literature on the population structure of S. aureus should be reviewed and included in this section.

The sampling was robust as authors sampled 5 buffaloes in 11 farms from 35 districts but it's not so clear how the farms or the number of buffaloes to be sampled from each farm was determined. If this was a convenience/random sampling, the authors should state this in the manuscript or better still give details on their calculation of the sample size.

The authors characterized the S. aureus using PFGE and analyzed the results using BioNumerics with a position tolerance of 1.5 as well as an optimization of 0. Are these parameters previously described? If yes, the author should cite appropriate references and if otherwise, a justification for using these parameters should be stated in the manuscript.

The authors described a high prevalence of staphylococci in presumably infected milk samples. How many samples carried both S. aureus and other staphylococci or non-staphylococci. Is there co-occurrence of the bacteria in different samples? If yes, is the co-occurrence specific, common or differ among water buffaloes from different districts? What are the identities of the non-staphylococcal isolates described in this study? As I understood, all the isolates are Gram positive and ferment mannitol as well.

The paragraph that addressed molecular characterization in the methods section is quite difficult to follow. The authors should consider splitting this paragraph into two. One for PCR-based and the other for PFGE. In the latter, the authors should consider stating the conditions and other information important for reproducing the results.

The authors stated that PCR assay that differentiates S. aureus from coagulase negative staphylococci, while also distinguishing MRSA from MSSA was used. The authors should consider giving a brief detail about this method and the basis for the differentiation, stating the exact gene(s) targeted by the assay. Although one could assume that nuc and mecA genes are targeted but this should be stated in the manuscript. How was the MLST determined?

The authors reported that the left quarters were twice prone to infection than those on the right side. Can the comment on this observation and its significance to the study? Again, is this district or farm associated?

t8934-ST22-IVa MRSA-IVa was isolated from all the districts except upper Kohistan. What is the prevalence of this clone in health care and community settings in the division studied? Has this clone been reported in animals/livestock or humans in the region? This information is important to contextualize this study.

There were also instances of two animals carrying two different clones (ST9-t7867-MSSA and ST101-t2078-MSSA), (ST9-t7867-MSSA and ST22-t8934-MRSA-IVa), (ST9-t7867-MSSA and ST9-MSSA) etc. Are these animals from the same farm? Also, were the two strains recovered from the same quarter? This information is also important.

The authors reported that there was no link between antibiotic resistance and the MSSA strain type, region, or farms. Is there a link between antibiotic resistance and MRSA and the farms or regions?

In the discussion section, the authors stated that studies on staphylococcal strains causing buffalo mastitis within Pakistan is scarce or not available. Are there studies from other food producing animals on this subject? Also, the authors conclude from their data that there is very limited diversity in S. aureus circulating amongst buffalo farms in this region of Pakistan. Are there animal exchange scenario or workers mobility between the farms studied? This information will be important to further support the conclusion. The authors should consider commenting on how homogenous climate affect the diversity of S. aureus.

Reviewer #2: The authors have produced a paper that clearly answers the research question posed. Also, the manuscript was comprehensive and well thought out. This work, on the epidemiology and molecular characterisation of S. aureus in bovine mastitis in Pakistan, is a very interesting addition to the literature where globally mastitis has a significant economic impact. It was also interesting to find that the predominant S. aureus CC, found in water buffalo mastitis, was not CC97 or CC151, the most frequently seen CC in cattle mastitis, but the livestock associated strain, CC9, commonly isolated from pigs in Asia.

6. PLOS authors have the option to publish the peer review history of their article (what does this mean?). If published, this will include your full peer review and any attached files.

Reviewer #1: No

Reviewer #2: **Yes: **Andrew Robb

---

## [Author Response · Author response to Decision Letter 0]

6 Apr 2022

Re: PONE-D-22-02764 Response to reviewers

Reviewer #1: 

Javed et al described the epidemiology and the prevalence of staphylococci causing bovine mastitis in water buffaloes in Hazara division of Pakistan. The authors used PFGE to assess the genetic relatedness of the S. aureus isolates recovered in the study and assessed them for antibiotic resistance. The study is well designed and should add to the growing data and knowledge of MRSA in food production animals in the region

Below are some comments for consideration

Minor comments

CoNS was not defined anywhere in the manuscript

-Our apologies, this was a formatting error. The section discussing CoNS was not supposed to be included in the manuscript and failed to get deleted. It has now been removed.

Except when starting a sentence, “staphylococci” or “staphylococcal” should not be written as “Staphylococci”

-The errors have been corrected.

Authors should consider reviewing the punctuations the manuscript.

-A review was done and changes made.

Major comments

1. The introduction is well written and very easy to read. It gave a detailed background of the study and identified the knowledge gaps this study aimed to fill. I would not want to assume that there is no single study from Pakistan that report the molecular characterization either from human, animals, or environmental matrices. Literature on the population structure of S. aureus should be reviewed and included in this section.

-Sorry, you are correct and the sentence was worded poorly. It has been changed to say: 

“While numerous studies have examined the prevalence and epidemiology of mastitis in various regions of Pakistan, detailed molecular typing of the staphylococci from cases of mastitis in the country is limited.” 

The population structure of S. aureus in humans in the region is presented (and has been expanded slightly) in the discussion, as it relates to the results. It truly is very limited with a couple studies only going as far as detecting the mecA gene. We feel that adding it to the introduction, specifically at the point where we are presenting the purpose and goals of the paper, would be distracting and dilute out the purpose of the paper. But we agree that the information deserves to be addressed in the paper. 

2. The sampling was robust as authors sampled 5 buffaloes in 11 farms from 35 districts but it's not so clear how the farms or the number of buffaloes to be sampled from each farm was determined. If this was a convenience/random sampling, the authors should state this in the manuscript or better still give details on their calculation of the sample size.

-The farms were randomly selected with the criteria that they be at least 5 Km apart and contain at least 5 animals. Animals were conveniently/randomly selected for sampling. This has been added to the materials and methods. A reference was included for how sampling (including numbers) was done. 

3. The authors characterized the S. aureus using PFGE and analyzed the results using BioNumerics with a position tolerance of 1.5 as well as an optimization of 0. Are these parameters previously described? If yes, the author should cite appropriate references and if otherwise, a justification for using these parameters should be stated in the manuscript.

-The reference is the same as the one describing the standardized protocol for PFGE typing S. aureus. A brief description was added to the section. 

4. The authors described a high prevalence of staphylococci in presumably infected milk samples. How many samples carried both S. aureus and other staphylococci or non-staphylococci. Is there co-occurrence of the bacteria in different samples? If yes, is the co-occurrence specific, common or differ among water buffaloes from different districts? What are the identities of the non-staphylococcal isolates described in this study? As I understood, all the isolates are Gram positive and ferment mannitol as well.

-The number of animals co-infected with SA and CoNS has been added to the manuscript and Figure 1. Co-infection with other species was not specifically tracked since only species that grew on our selective media would be accounted for, creating an inaccurate number. 

5. The paragraph that addressed molecular characterization in the methods section is quite difficult to follow. The authors should consider splitting this paragraph into two. One for PCR-based and the other for PFGE. In the latter, the authors should consider stating the conditions and other information important for reproducing the results.

-The information was presented in the current order because it provides the easiest workflow for anyone wanting to repeat the experiments. The multiplex PCR is routinely performed first for a quick identification of species and methicillin resistance. PFGE is done next on all S. aureus to identify how many strain types/groups are likely present. Antiseptic resistance PCR, SCCmec typing, agr typing, spa typing and MLST typing are subsequently done for greater detail and strain molecular characterization. Grouping them based on PCR vs PFGE would create a workflow that isn’t ideal. In terms of the PFGE conditions, they are described in the references, however the following information was added: 

“S. aureus isolates were subsequently fingerprinted using pulsed field gel electrophoresis (PFGE) following digestion by SmaI, using a standardized protocols [11]. Briefly, S. aureus agarose plugs were digested with smaI and loaded onto a 1% agarose gel, then electrophoresis done with a CHEF mapper using switch times of 5.3 to 35 for 18 hours at 14°C, 6.0V/cm with an angle of 120 in 0.5x TBE.”

6. The authors stated that PCR assay that differentiates S. aureus from coagulase negative staphylococci, while also distinguishing MRSA from MSSA was used. The authors should consider giving a brief detail about this method and the basis for the differentiation, stating the exact gene(s) targeted by the assay. Although one could assume that nuc and mecA genes are targeted but this should be stated in the manuscript. How was the MLST determined?

-Detailed information is available in the reference however a brief mention of the genes used in the multiplex has been added to the text. We apologize about the MLST omission (formatting error again), the reference was added back. 

7. The authors reported that the left quarters were twice prone to infection than those on the right side. Can the comment on this observation and its significance to the study? Again, is this district or farm associated?

-The topic was addressed in lines 331-346 of the discussion (original submission), with a slight edit in this version. There is no association between district or farm, which was added to the discussion section.

“We observed that left quarters (both front and rear) had more than twice the infection rate (69.6%) of right quarters (30.4%), which held true in all districts and on all farms.”

8. t8934-ST22-IVa MRSA-IVa was isolated from all the districts except upper Kohistan. What is the prevalence of this clone in health care and community settings in the division studied? Has this clone been reported in animals/livestock or humans in the region? This information is important to contextualize this study.

-This was addressed in the discussion in lines 369-370 (original submission). With such limited molecular data available for Pakistan, the actual prevalence of ST22-MRSA in health care and community settings is unknown, both for the country as a whole and division studied. There was, likewise, no data from other animals/livestock. 

“In one study CC22-MRSA-IV was identified in 13.6% (belonging to three different strains) of the human isolates [33], matching the ST type of all the MRSA in our study.”

9. There were also instances of two animals carrying two different clones (ST9-t7867-MSSA and ST101-t2078-MSSA), (ST9-t7867-MSSA and ST22-t8934-MRSA-IVa), (ST9-t7867-MSSA and ST9-MSSA) etc. Are these animals from the same farm? Also, were the two strains recovered from the same quarter? This information is also important.

-No they did not occur in the same farm, as previously stated in the results on lines 247-248 (original submission). They were not in the same quarter either, so the sentence has been amended to read:

“Interestingly, there were ten instances where different strain types were isolated from a single animal, with each strain isolated from different quarters within the animal, and all occurring on different farms.”

10. The authors reported that there was no link between antibiotic resistance and the MSSA strain type, region, or farms. Is there a link between antibiotic resistance and MRSA and the farms or regions?

- We added that MRSA showed no link either. Now read:

“There wasn’t a link between antibiotic resistance and the MSSA/MRSA strain type, region or farm from which the samples came (details available in S1 Table).”

11. In the discussion section, the authors stated that studies on staphylococcal strains causing buffalo mastitis within Pakistan is scarce or not available. Are there studies from other food producing animals on this subject? 

-Unfortunately we found limited studies with detailed molecular data for S. aureus in other food or companion animals. We included what we could find in the discussion. These are examples of the level of detail typically reported in the region.

https://doi.org/10.1089/fpd.2018.2585: PFGE but not shown, mecA, mecC, antibiotic resistance in eggs of household chickens.

https://doi.org/10.3389/fmicb.2020.577707: detected mecA and spa amplicon size (not sequenced). In chickens, beef, mutton of butcher shops. 

http://dx.doi.org/10.17582/journal.pjz/2017.49.3.861.867: susceptibility testing. Camels.

http://dx.doi.org/10.19045/bspab.2021.100101: mecA PCR. Cats and dogs.

12. Also, the authors conclude from their data that there is very limited diversity in S. aureus circulating amongst buffalo farms in this region of Pakistan. Are there animal exchange scenario or workers mobility between the farms studied? This information will be important to further support the conclusion. The authors should consider commenting on how homogenous climate affect the diversity of S. aureus.

-Thank you for this great suggestion. We have add the following information in the:

“This suggests that there is very limited diversity in the S. aureus circulating amongst buffalo farms in this region of Pakistan, which could be due to the close geographic connection between farms and districts in the division, as well as their homogenous climate. With similar environmental factors (such as humidity and temperature) driving selection, similar strain types might prove to be the most fit in each district of this region. Additionally, the farms are small scale and just 5 km apart, meaning the same people could be visiting multiple farms to purchase milk, acting as sources of pathogen transfer. On top of that, the owners of farms in close proximity are often relatives and there can be an exchange of animals between farms, again being a source of pathogen transfer.”

Reviewer #2: 

The authors have produced a paper that clearly answers the research question posed. Also, the manuscript was comprehensive and well thought out. This work, on the epidemiology and molecular characterisation of S. aureus in bovine mastitis in Pakistan, is a very interesting addition to the literature where globally mastitis has a significant economic impact. It was also interesting to find that the predominant S. aureus CC, found in water buffalo mastitis, was not CC97 or CC151, the most frequently seen CC in cattle mastitis, but the livestock associated strain, CC9, commonly isolated from pigs in Asia.

-Thank you for your appreciation of our manuscript.

---

## [Decision Letter · Decision Letter 1]

20 Apr 2022

PONE-D-22-02764R1Epidemiology and Molecular Characterization of Staphylococcus aureus Causing Bovine Mastitis in Water Buffaloes from the Hazara Division of Khyber Pakhtunkhwa, PakistanPLOS ONE

Dear Dr. Zhang

Thank you for submitting your manuscript to PLOS ONE. After careful consideration, we feel that it has merit but does not fully meet PLOS ONE’s publication criteria as it currently stands. Therefore, we invite you to submit a revised version of the manuscript that addresses the additional points raised by reviewer #2.

We look forward to receiving your revised manuscript.

Kind regards,

Herminia de Lencastre, Ph.D.

Academic Editor

PLOS ONE

Journal Requirements:

Reviewers' comments:

Reviewer's Responses to Questions

**Comments to the Author**

1. If the authors have adequately addressed your comments raised in a previous round of review and you feel that this manuscript is now acceptable for publication, you may indicate that here to bypass the “Comments to the Author” section, enter your conflict of interest statement in the “Confidential to Editor” section, and submit your "Accept" recommendation.

Reviewer #1: All comments have been addressed

Reviewer #2: (No Response)

2. Is the manuscript technically sound, and do the data support the conclusions?

Reviewer #1: Yes

Reviewer #2: Yes

3. Has the statistical analysis been performed appropriately and rigorously? 

Reviewer #1: Yes

Reviewer #2: N/A

4. Have the authors made all data underlying the findings in their manuscript fully available?

Reviewer #1: Yes

Reviewer #2: Yes

5. Is the manuscript presented in an intelligible fashion and written in standard English?

Reviewer #1: Yes

Reviewer #2: Yes

6. Review Comments to the Author

Reviewer #1: The authors addressed all the comments and considered all the suggestions. I have no further comments.

Reviewer #2: I think that this publication has been well thought out and presented. However, there are a few very minor points.

Line 46-47 states "While numerous studies etc". This statement needs to be referenced.

Line 86 an should read and.

Line 94 gentamycin - spelling is gentamicin.

Line 96 Please state the company name of the antibiotic disk supplier.

Line 105 Switch times are in seconds.

Line 108 The author should differentiate between antiseptic resistance genes (qac) and antibiotic resistance genes (mup).

Line 126 34.55% SFMT positive but figure for antimicrobial growth on line 127 states 43.55%.

Line 130 the number 27 should be written as Twenty-seven as it begins the sentence

line 227-228 Table 5 again the author should differentiate antiseptic and antibiotic resistance genes

Line 360 "however, in one study characterized" would read better as "however, in a study that characterized"

Line 422 "With limited data currently available, this become crucial background" This sentence is not clear.

7. PLOS authors have the option to publish the peer review history of their article (what does this mean?). If published, this will include your full peer review and any attached files.

Reviewer #1: No

Reviewer #2: No

---

## [Author Response · Author response to Decision Letter 1]

22 Apr 2022

PONE-D-22-02764R1 Response to reviewers

Reviewer #1: 

The authors addressed all the comments and considered all the suggestions. I have no further comments.

Reviewer #2: 

I think that this publication has been well thought out and presented. However, there are a few very minor points.

Line 46-47 states "While numerous studies etc". This statement needs to be referenced.

-several references added

Line 86 an should read and.

-thank you, corrected

Line 94 gentamycin - spelling is gentamicin.

-corrected

Line 96 Please state the company name of the antibiotic disk supplier.

-added

Line 105 Switch times are in seconds.

-added

Line 108 The author should differentiate between antiseptic resistance genes (qac) and antibiotic resistance genes (mup).

-differentiated

Line 126 34.55% SFMT positive but figure for antimicrobial growth on line 127 states 43.55%.

-Sorry, corrected

Line 130 the number 27 should be written as Twenty-seven as it begins the sentence

-amended

line 227-228 Table 5 again the author should differentiate antiseptic and antibiotic resistance genes

-differentiated

Line 360 "however, in one study characterized" would read better as "however, in a study that characterized"

-sorry, the word was missed during final track change review.

Line 422 "With limited data currently available, this become crucial background" This sentence is not clear.

-Changed to: “With limited molecular data currently available, our study provides crucial background information for future researchers, and could help elucidate the epidemiological spread of a pathogen that causes significant economic impact on a major industry in the country.”

---

## [Decision Letter · Decision Letter 2]

25 Apr 2022

Epidemiology and Molecular Characterization of Staphylococcus aureus Causing Bovine Mastitis in Water Buffaloes from the Hazara Division of Khyber Pakhtunkhwa, Pakistan

PONE-D-22-02764R2

Dear Dr. Zhang,

We’re pleased to inform you that your manuscript has been judged scientifically suitable for publication and will be formally accepted for publication once it meets all outstanding technical requirements.

Kind regards,

Herminia de Lencastre, Ph.D.

Academic Editor

PLOS ONE

Additional Editor Comments (optional):

Reviewers' comments:

Reviewer's Responses to Questions

**Comments to the Author**

1. If the authors have adequately addressed your comments raised in a previous round of review and you feel that this manuscript is now acceptable for publication, you may indicate that here to bypass the “Comments to the Author” section, enter your conflict of interest statement in the “Confidential to Editor” section, and submit your "Accept" recommendation.

Reviewer #2: All comments have been addressed

2. Is the manuscript technically sound, and do the data support the conclusions?

Reviewer #2: Yes

3. Has the statistical analysis been performed appropriately and rigorously? 

Reviewer #2: N/A

4. Have the authors made all data underlying the findings in their manuscript fully available?

Reviewer #2: Yes

5. Is the manuscript presented in an intelligible fashion and written in standard English?

Reviewer #2: Yes

6. Review Comments to the Author

Reviewer #2: (No Response)

7. PLOS authors have the option to publish the peer review history of their article (what does this mean?). If published, this will include your full peer review and any attached files.

Reviewer #2: **Yes: **Dr Andrew Robb

---

## [Editor Report · Acceptance letter]

28 Apr 2022

PONE-D-22-02764R2 

Epidemiology and Molecular Characterization of *Staphylococcus aureus* Causing Bovine Mastitis in Water Buffaloes from the Hazara Division of Khyber Pakhtunkhwa, Pakistan 

Dear Dr. Zhang:

I'm pleased to inform you that your manuscript has been deemed suitable for publication in PLOS ONE. Congratulations! Your manuscript is now with our production department. 

Kind regards, 

on behalf of

Dr. Herminia de Lencastre 

Academic Editor

PLOS ONE